# Evaluation of Metabarcoding Primers for Analysis of Soil Nematode Communities

**Md. Maniruzzaman Sikder [1,2], Mette Vestergård [1], Rumakanta Sapkota [3], Tina Kyndt [4] and Mogens Nicolaisen [1,*]**

[1]  Department of Agroecology, Faculty of Technical Sciences, Aarhus University, 4200 Slagelse, Denmark; mms@agro.au.dk (M.M.S.); mvestergard@agro.au.dk (M.V.)

[2]  Department of Botany, Faculty of Biological Sciences, Jahangirnagar University, Savar 1342, Dhaka, Bangladesh

[3]  Department of Environmental Science, Faculty of Technical Sciences, Aarhus University, 4000 Roskilde, Denmark; rs@envs.au.dk

[4]  Department of Molecular Biotechnology, Faculty of Bioscience Engineering, Ghent University, 9000 Gent, Belgium; tina.kyndt@ugent.be

*  Correspondence: mn@agro.au.dk; Tel.: +4524757668

**Abstract:** While recent advances in next-generation sequencing technologies have accelerated research in microbial ecology, the application of high throughput approaches to study the ecology of nematodes remains unresolved due to several issues, e.g., whether to include an initial nematode extraction step or not, the lack of consensus on the best performing primer combination, and the absence of a curated nematode reference database. The objective of this method development study was to compare different primer sets to identify the most suitable primer set for the metabarcoding of nematodes without initial nematode extraction. We tested four primer sets for amplicon sequencing: JB3/JB5 (mitochondrial, I3-M11 partition of COI gene), SSU_04F/SSU_22R (18S rRNA, V1-V2 regions), and Nemf/18Sr2b (18S rRNA, V6-V8 regions) from earlier studies, as well as MMSF/MMSR (18S rRNA, V4-V5 regions), a newly developed primer set. We used DNA from 22 nematode taxa, 10 mock communities, 20 soil samples, 4 root samples, and one bulk soil. We amplified the target regions from the DNA samples with the four different primer combinations and sequenced the amplicons on an Illumina MiSeq sequencing platform. We found that the Nemf/18Sr2b primer set was superior for detecting soil nematodes compared to the other primer sets based on our sequencing results and on the annotation of our sequence reads at the genus and species ranks. This primer set generated 74% reads of Nematoda origin in the soil samples. Additionally, this primer set did well with the mock communities, detecting all the included specimens. It also worked better in the root samples than the other primer set that was tested. Therefore, we suggest that the Nemf/18Sr2b primer set could be used to study rhizosphere soil and root associated nematodes, and this can be done without an initial nematode extraction step.

**Keywords:** nematode diversity; soil; rhizosphere; environmental; NGS

## 1. Introduction

Nematodes are highly diverse and abundant metazoans with a worldwide distribution [1,2]. Generally, nematologists have relied on classical morphology-based taxonomy along with biochemical or molecular methods for nematode identification [3,4]. Morphological identification is difficult, requires taxonomic expertise [5], and often becomes challenging when it comes to identifying nematodes at lower taxonomic levels [6]. DNA-based identification has eased the task of taxonomic nematode identification in recent years [7–10].

Initially, the barcoding approach was introduced for species-level detection [11–13]. The mitochondrial cytochrome oxidase I gene (COI gene) has been successfully used as barcode for the identification of nematodes and for resolving taxonomic relationships among closely related species [14–16]. The COI gene has been shown to provide a greater taxonomic resolution than the small subunit (18S rRNA) rDNA [17]. The potential of COI gene-based barcoding specific to particular groups of nematodes has been explored on pure DNA samples of various nematode taxa including root-knot nematodes [18], marine nematodes [16], Aphelenchoididae [19] and *Pratylenchus* [20]. A lack of suitable COI-based consensus primers for the detection of nematode diversity in comparison to 18S-based primers limits the implementation of COI-based metabarcoding for nematodes [21]. Though metabarcoding has been in use for more than a decade [22–24], this method is still under development and standardization for nematode community analysis [21,25–28] due to a lack of reliable quantification methods, PCR biases [29,30], differences in rRNA gene copy numbers among nematode species [31], limited reference databases [21,32], and a lack of consensus on primer sets [21,25,27].

The 18S rRNA gene has been reported to evolve more conservatively than the COI gene and is often used to distinguish between nematode species that are not closely related [33,34]. Though 18S rRNA lacks power for the discrimination of closely related nematode species, the presence of conserved regions in the 18S rRNA gene has allowed for the design of several promising primer sets for the generic amplification of a wide diversity of nematodes [23,25,35]. The V2, V4, and V9 variable regions of the 18S rRNA gene were suggested to be most suitable for biodiversity assessments after multiple alignment of eukaryotic 18S rRNA gene sequences available in the SILVA database [36]. Consequently, the 18S rRNA gene may remain the most widely used molecular marker for the identification of nematodes in high throughput sequencing (HTS) approaches [21,37]. An 18S rRNA gene-based HTS approach was proposed as a promising solution for the description of nematode communities from DNA extracted from environmental samples [23]. Likewise, several nematode metabarcoding primer combinations have been designed based on the variable regions of the 18S rRNA gene and used to detect soil nematodes extracted from samples [21,25,27]. However, these primer sets were tested on nematodes extracted from soil and not on DNA extracted directly from soil samples. We aimed for a suitable primer set that would amplify and detect soil nematodes from environmental DNA without prior nematode extraction from soil, as well as in the presence of non-target plant, fungal, protist, and metazoan DNA. Therefore, the aim of the present method development study was to compare commonly used primer sets from the literature and a newly designed primer set to identify the most suitable primer set for the metabarcoding of plant-parasitic and free-living soil nematodes. For this, we used DNA extracted from 22 different nematode species, 10 mock communities in water, 20 DNA samples extracted from agricultural fields, 4 root samples with attached soil, and a bulk soil to validate the primer sets.

## 2. Materials and Methods

### 2.1. Primer Sets

We selected four primer sets for the amplicon sequencing of nematode taxa (Table S1 and Figure S1). The primer set SSU_04F/SSU_22R (SSU) amplifies the V1-V2 regions of the 18S rRNA gene and was recently used to describe the assemblages of free-living soil nematodes after nematode isolation from soil using the MiSeq platform [21,26,27]. We designed a primer set, MMS (MMSF: 5′-GGTGCCAGCAGCCGCGGTA-3′, MMSR: 5′-CTTTAAGTTTCAGCTTTGC-3′) located in the variable

V4-V5 regions of the 18S rRNA gene. Furthermore, we included the Nemf/18Sr2b primer set (NEM), which was developed for the 454-sequencing platform using a semi-nested PCR approach. In the present study, the second PCR step of the nested PCR was omitted, which resulted in a larger PCR product (around 500 bp) covering the V6-V8 regions of the 18S rRNA gene. We also tested the JB3/JB5 (JB)-targeting of the I3-M11 region of the mitochondrial COI gene in the present study; this primer set was previously used to study nematode communities after initial nematode isolation [17,21,25]. Several other mitochondrial primer sets, i.e., COX2F/COX2R, COX3F/COX3R [18], JB2/JB5GED [38], and COIF/COIR [39], were initially tested on individual nematode taxa, but due to very low PCR amplification success rates, these primer sets were not included in the subsequent analyses.

### 2.2. Nematode Species, Mock Communities and Soil Samples

In order to test the primer sets, we used DNA extracts from 22 soil nematode species kindly provided by researchers (Table S2). We included plant-parasitic nematodes representing several agriculturally important nematode families (Meloidogynidae, Pratylenchidae, Heteroderidae, and Dolichodoridae) and a free-living nematode taxon belonging to Rhabditidae. We also tested the primer sets on ten mock communities in which the nematode DNA of each specimen was pooled on a volumetric basis (2.5 µL DNA of each taxon) (Table S3).

We used 20 soil samples collected from different fields, as described earlier [40] (Table S4). To represent each soil sample, 20 randomly distributed sites in the field were selected, and 200 g of upper soil (15 cm soil layer) of each site was sampled and thoroughly mixed by hand before 100 g subsamples were taken. Samples were freeze-dried for 3 days and subsequently ground in a mixer mill (Retsch MM301, Haan, Germany) for 10 min. We used 0.25 g of soil for DNA extraction from the thoroughly ground and homogenized 100 g soil samples.

### 2.3. Root Samples

We grew 4 different *Arabidopsis thaliana* lines (N6567, N8034, N929, and NW25) to test primer efficiency in detecting soil nematodes in roots (including attached soil). We filled 20 pots each with 400 g of homogenized clayey sandy soil (pH = 5.9), and we transplanted three-week old 5 *Arabidopsis* seedlings into each pot. Four pots were filled with soil and kept without plants—these were considered bulk soil (BS). Pots were placed randomly in a glasshouse at 22–24 °C and 12 h/12 h light–dark conditions. After 5 weeks, roots were harvested by gently pressing the sides of pots to loosen soils around the plant before careful uprooting. We also took a bulk soil sample (10 g) with 4 biological replicates. Roots were shaken gently to remove loosely adhering soil. The five plant roots from each pot were pooled to represent a biological replicate. Four biological replicates of each *Arabidopsis* line were harvested. The roots were quickly transferred to collecting tubes and plunged into liquid nitrogen. All samples were transferred to a −80 °C freezer until further use. Before DNA extraction, bulk soil and root samples were freeze-dried before grinding in a Geno/Grinder 2000 (RAMCON, Denmark) at 1500 rpm for 6 × 1 min.

### 2.4. DNA Extraction, PCR and Sequencing Library Preparation

DNA was extracted from 0.25 g of each of the ground soil and root samples using the PowerLyzer soil DNA extraction kit (Qiagen, Hilden, Germany) according to the manufacturer's instructions, except that samples were homogenized in a Geno/Grinder 2000 (RAMCON, Denmark) at 1500 rpm for 3 × 30 s. DNA concentrations were measured using a Qubit Fluorometer (Thermo Fisher Scientific, Waltham, Massachusetts, USA) and diluted to 2 ng/µL in each soil and root sample.

To amplify target regions, PCR was performed in a 25 µL reaction mixture consisting of 5 µL of a Promega 5X reaction buffer (Promega Corporation, Madison, USA), 1.5 µL of $MgCl_2$ (25 mM), 2 µl of dNTPs (2.5 mM), 0.5 µL of each primer (10 µM), 0.125 µL of GoTaq Flexi polymerase (5U, Promega Corporation, Madison, USA), and 2 µL of a DNA template (1 ng/µL for individual nematode species and mock communities; 2 ng/µL for soil and root samples). All four primer sets were tested on individual taxa, mock, and soil samples. In addition, the MMS and NEM primer sets were used on

root samples. PCR cycles for the JB primer combination were 94 °C for 5 min (94 °C for 1 min, 50 °C for 30 s, and 72 °C for 45 s) 35 cycles, 72 °C for 10 min, and 4 °C on hold [16]. Similar PCR cycles were used for the other sets, except that the annealing temperature was set to 53 °C for MMS and NEM and set to 55 °C for the SSU primer set [41]. Each of the primer sets of the first PCR (Table S1) were tagged with the Illumina adapter overhang nucleotide sequence. PCR products of the first PCR were diluted (1:5) and pooled for each sample.

A second PCR was performed for dual indexing to enable the pooling of all samples. The master mix of this PCR was identical to the first PCR except that 2 μL of the DNA template and 2 μL of the index primer combinations were used. Each index primer consisted of a specific sequence for Illumina sequencing, a unique 8 bp multiplex identifier, and the Illumina adapter overhang sequence. The second PCR was performed with the following cycles: 94 °C for 5 min, (94 °C for 30 s, 55 °C for 30 s, and 72 °C for 1 min) 13 cycles, 72 °C for 10 min, and 4 °C on hold. All amplicons were visualized by gel electrophoresis, pooled, precipitated, and the pellet dissolved in DNAse free water. Pooled DNA was run on a gel, and amplicons were excised and purified using the QIAquick Gel Extraction kit (Qiagen, Hilden, Germany) according the manufacturer's instructions. Finally, the DNA concentrations were measured by Qubit Fluorometer (Thermo Fisher Scientific, Waltham, Massachusetts, USA), and DNA libraries were sent for sequencing on an Illumina MiSeq sequencer with PE300 (Eurofins Genomics, Ebersberg, Germany).

### 2.5. Sequence Analysis

Sequences obtained from the Illumina MiSeq run were demultiplexed using a demultiplexer [42]. In brief, paired-end reads were joined using VSEARCH version 2.6 of QIIME2 [43]. To join the paired-end reads, we used a default overlapping minimum read length of 10 base pairs and removed reads with quality Phred scores of <30. Internal barcodes, forward and reverse primers, and reads less than 250 base pairs were also excluded. Following this, sequences were dereplicated and screened for chimera detection and clustered at a 99% similarity level using VSEARCH version 2.6. Taxonomy assignments for the clustered operational taxonomic units (OTUs) were done in the SILVA 132 reference database for 18S [44,45] and the MIDORI curated database for mitochondrial-encoded sequence reads using assign_taxonomy.py [46]. Moreover, all nematode OTUs were queried against the NCBI GenBank database to reconfirm their taxonomic assignments.

### 2.6. Statistical Analysis

Community composition was determined using OTUs from a bulk soil and root samples. We used phyloseq R packages for diversity-based calculations [47]. The OTU table was transformed before the calculation of relative abundance and rarified before OTU richness and alpha diversity measure determination. Beta diversity-based dissimilarities matrices on nematode communities for the partitioning of variance was calculated using PERMANOVA and Adonis test in R statistics.

## 3. Results

### 3.1. Data Characteristics

We analyzed sequence reads from the 22 nematodes taxa, 10 mock communities and 20 soils using the four primer sets. After quality control, sequence reads were clustered into 95, 534, 273, and 258 Nematoda OTUs for JB (JB3/JB5), SSU (SSU_04F/SSU_22R), MMS (MMSF/MMSR), and NEM (Nemf/18Sr2b) primer sets, respectively. The sequence reads of root samples were clustered into 58 and 126 Nematoda OTUs for the MMS and NEM primer sets, respectively.

### 3.2. Individual Nematode Species

For the individual nematode species, we could assign 8 of the 22 species to species rank and 10 to genus rank using the JB primer set, whereas four species were not amplified (Table 1). Using the SSU primer set, only 12 out of the 22 species were amplified, of which seven were assigned to genus

rank and five to species rank. The MMS primer set amplified all nematodes species except for *Meloidogyne graminicola* (Table 1). This primer set identified *Meloidogyne* at the genus rank except for *Meloidogyne hapla,* which was assigned at the species rank. Five nematode taxa were also assigned at the species rank. The NEM primer set successfully amplified all the root knot nematode species except for *M. graminicola.* Most *Meloidogyne* species were assigned to the genus rank except for *Meloidogyne hapla*, which was assigned to the species level. Four nematode taxa were assigned at the species rank by the NEM primer set. Cyst nematodes (*Heterodera carotae* and *Heterodera schachtii*) were assigned to the family rank with the NEM primer set (Table 1). The root lesion nematodes *Pratylenchus* spp. were detected as *Pratylenchus neglectus* and *Pratylenchus penetrans* by both the JB and NEM primer sets; MMS detected these nematodes as *Pratylenchus neglectus* and *Pratylenchus* sp., while the SSU primer set did not amplify root lesion nematodes.

**Table 1.** The efficiency of four metabarcoding primer sets in the detection of individual nematode taxa at different taxonomic ranks using Illumina sequencing.

| Nematode Taxa | JB | SSU | MMS | NEM |
|---|---|---|---|---|
| *Meloidogyne incognita* | Genus | Not detected | Genus | Genus |
| *Meloidogyne arenaria* | Genus | Genus | Genus | Genus |
| *Meloidogyne javanica* | Genus | Genus | Genus | Genus |
| *Meloidogyne graminicola* | Genus | Not detected | Not detected | Not detected |
| *Meloidogyne ethiopica* | Genus | Genus | Genus | Genus |
| *Meloidogyne inornata* | Genus | Genus | Genus | Genus |
| *Meloidogyne ulmi* | Genus | Genus | Genus | Genus |
| *Meloidogyne luci* | Genus | Genus | Genus | Genus |
| *Meloidogyne hapla* | Species | Genus | Species | Species |
| *Meloidogyne enterolobii* | Species | Not detected | Genus | Genus |
| *Meloidogyne chitwoodi* | Species | Not detected | Genus | Genus |
| *Meloidogyne fallax* | Genus | Not detected | Genus | Genus |
| *Meloidogyne minor* | Genus | Species | Genus | Genus |
| *Meloidogyne naasi* | Species | Not detected | Genus | Genus |
| *Pratylenchus penetrans* | Species | Not detected | Genus | Species |
| *Pratylenchus spp.* | Species | Not detected | Species | Species |
| *Heterodera carotae* | Not detected | Not detected | Genus | Family |
| *Heterodera schachtii* | Species | Species | Genus | Family |
| *Belonolaimus longicaudatus* | Not detected | Species | Species | Species |
| *Bursaphelenchus mucronatus* | Not detected | Not detected | Genus | Genus |
| *Caenorhabditis elegans* | Species | Species | Species | Species |
| *Ditylenchus dipsaci* | Not detected | Species | Species | Genus |

The NCBI Blast tool was used for taxonomic assignments, and top hits with 100% coverage, and 100% sequence similarities at the species rank and ≥99% at the genus rank, were considered. JB: JB3/JB5; SSU: SSU_04F/SSU_22R; MMS: MMSF/MMSR; NEM: Nemf/18Sr2b.

### 3.3. Mock Communities

In mock communities, JB-generated OTUs were assigned to the genus rank within the Meloidogynidae family, except for *Meloidogyne hapla* and *Meloidogyne naasi*, which were assigned to the species rank, and *Meloidogyne minor*, which was not amplified (Table 2). The cyst nematode taxa *Globodera pallida*, *Globodera rostochiensis,* and *Heterodera schachtii* were detected at the species rank, while *Heterodera carotae* was not amplified. Taxa within Pratylenchidae and Rhabditidae were assigned at the species rank, and *Belonolaimus longicaudatus* was not detected (Table 2).

The SSU-generated OTUs were assigned to the genus rank within the Meloidogynidae, except for three taxa (*Meloidogyne minor, Meloidogyne hapla*, and *Meloidogyne inornata*) that were not amplified. Three taxa within Heteroderidae were detected at the genus rank, while *Globodera pallida* was assigned to the species rank. *Caenorhabditis elegans* was also assigned to the species rank. The SSU primers failed to amplify *Pratylenchus penetrans* and *Belonolaimus longicaudatus* in any mock community sample (Table 2).

The MMS-generated OTUs from ten taxa of the Meloidogynidae were assigned to the genus rank, while *Meloidogyne hapla* was assigned to the species rank and *Pratylenchus penetrans* was not detected in mock communities. The OTUs belonging to the taxa *Globodera pallida, Heterodera schachtii*, and *Caenorhabditis elegans* were assigned to the species rank (Table 2).

The NEM primer set amplified and detected nematode taxa of Meloidogynidae in accordance with the MMS primer pair; *Pratylenchus penetrans, Belonolaimus longicaudatus*, and *Caenorhabditis elegans* were assigned at the species level; the *Globodera* species were assigned at the genus rank; and the two *Heterodera* species were assigned at the family rank (Table 2).

**Table 2.** The efficiency of four metabarcoding primers in the detection of taxa in mock communities using Illumina sequencing.

| Taxa in Mock Communities | JB | SSU | MMS | NEM |
|---|---|---|---|---|
| *Meloidogyne incognita* | Genus | Genus | Genus | Genus |
| *Meloidogyne arenaria* | Genus | Genus | Genus | Genus |
| *Meloidogyne ethiopica* | Genus | Genus | Genus | Genus |
| *Meloidogyne inornata* | Genus | Not detected | Genus | Genus |
| *Meloidogyne ulmi* | Genus | Genus | Genus | Genus |
| *Meloidogyne luci* | Genus | Genus | Genus | Genus |
| *Meloidogyne hapla* | Species | Not detected | Species | Species |
| *Meloidogyne chitwoodi* | Genus | Genus | Genus | Genus |
| *Meloidogyne fallax* | Genus | Genus | Genus | Genus |
| *Meloidogyne minor* | Not detected | Not detected | Genus | Genus |
| *Meloidogyne naasi* | Species | Genus | Genus | Genus |
| *Pratylenchus penetrans* | Species | Not detected | Not detected | Species |
| *Heterodera carotae* | Not detected | Genus | Genus | Family |
| *Heterodera schachtii* | Species | Genus | Species | Family |
| *Belonolaimus longicaudatus* | Not detected | Not detected | Species | Species |
| *Caenorhabditis elegans* | Species | Species | Species | Species |
| *Globodera pallida* | Species | Species | Species | Genus |
| *Globodera rostochiensis* | Species | Genus | Genus | Genus |

The NCBI Blast tool was used for taxonomic assignments, and top hits with 100% coverage, as well as 100% sequence similarities at the species rank and ≥99% at the genus rank, were considered. JB: JB3/JB5; SSU: SSU_04F/SSU_22R; MMS: MMSF/MMSR; NEM: Nemf/18Sr2b.

### 3.4. Detection of Nematodes in Soil Samples

For the JB primer set, only 4% of the total sequence reads from the soil samples were classified as Nematoda (Figure 1). For the SSU primer set, only 1% of the sequence reads from the soil samples were classified as Nematoda (Figure 1). This primer set amplified a broad spectrum of other eukaryotes such as fungi, plant, Cercozoa, and Charophyta. For the newly designed primer set (MMS), 14% of total sequence reads belonged to Nematoda, and for the NEM primer set, 74% of the total sequence reads were assigned to Nematoda in the soil samples (Figure 1).

There was a low recovery of sequence reads that assigned to Nematoda from soil samples using the JB and SSU primer sets (Figure 1, Figure S2, and Table S5). In contrast, the NEM and MMS primers detected a wide range of nematode taxa from different families from the 0.25 g soil samples (Table S5). Twenty nematode families were detected using both the NEM and MMS primer sets, and seven families were uniquely detected by each primer set (Figure 2A). We recorded 13 and 12 different

unique genera in soil samples by the NEM and MMS primer sets, respectively, and 25 genera were detected by both primer sets (Figure 2A). We detected 29 and 20 unique nematode species with the NEM and MMS primer sets, respectively, while 23 species were detected by both primer sets (Figure 2A). These two primers sets were able to detect a high diversity of nematode taxa in the soil samples compared to the JB and SSU primer sets (Figures S2 and S3).

### 3.5. Nematode Communities in Root Samples

Next, we tested the efficiencies of the NEM and MMS primers in the amplification and detection of nematodes associated with *Arabidopsis* roots and a bulk soil sample. Due to the low performance of the JB and SSU primer sets in the above analyses, we did not include these primers in the subsequent analysis of primer performance in the soil and root samples. In the root samples, we recovered 30 families, 35 genera, and 51 species with the NEM primers, whereas the MMS primers detected 15 families, 21 genera, and 31 species (Figure 2B). All 15 families detected by the MMS primers were also detected by the NEM primers, whereas 15 unique families were only detected with the NEM primers (Figure 2B). Sixteen genera were uniquely detected by NEM, whereas only two genera were unique to MMS and 19 genera were detected by both primers sets. We detected 28 unique nematode species with the NEM primer set, while only eight unique species were detected by the MMS primer set and 23 species were detected by both primer sets (Figure 2B).

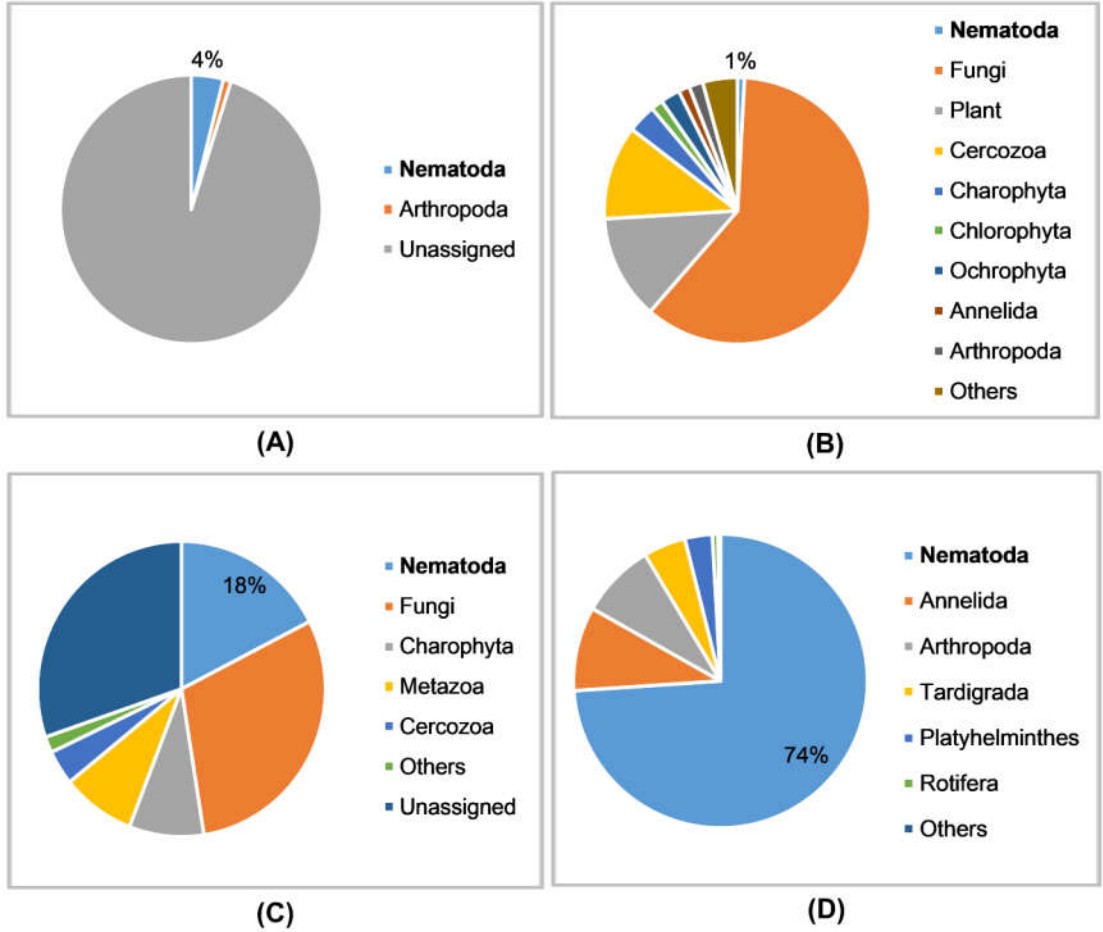

**Figure 1.** Relative distribution of sequence reads in soil samples amplified with JB: JB3/JB5 (**A**), SSU: SSU_04F/SSU_22R (**B**), MMS: MMSF/MMSR (**C**), and NEM: Nemf/18Sr2b (**D**) primer sets; percentage indicates the proportion of sequence reads that were assigned to Nematoda.

Both primer sets detected plant-parasitic and free-living nematode taxa in the bulk soil and root samples (Figure 3). The NEM primer set detected a higher number of nematode taxa compared to the MMS primer set (Table S6). The relative abundance of two important plant-parasitic taxa *Meloidogyne* and *Pratylenchus* in root samples were comparatively lower when using the MMS primer set compared to the NEM primer set (Figure 3A,B). Two taxa, *Mesorhabditis* and *Oscheius,* belonging to Rhabditidae appeared among the ten most abundant taxa in MMS dataset, whereas these taxa were not found among the ten most abundant genera in the NEM dataset (Figure 3A,B). However, the NEM primer set was able to detect *Mesorhabditis belari*, and *Oscheius tipulae* in root samples (Table S6). The dataset generated by the NEM primer set had a higher observed OTU richness and Shannon diversity indices compared to the MMS primer set (Figure 3C,D). The NEM primer set explained 68% of the variation of nematode communities ($p < 0.001$), whereas 59% of the variation ($p < 0.001$) was explained by the MMS primer set in Principal Coordinates Analysis (PCoA) plots (Figure 3E,F and Tables S7 and S8).

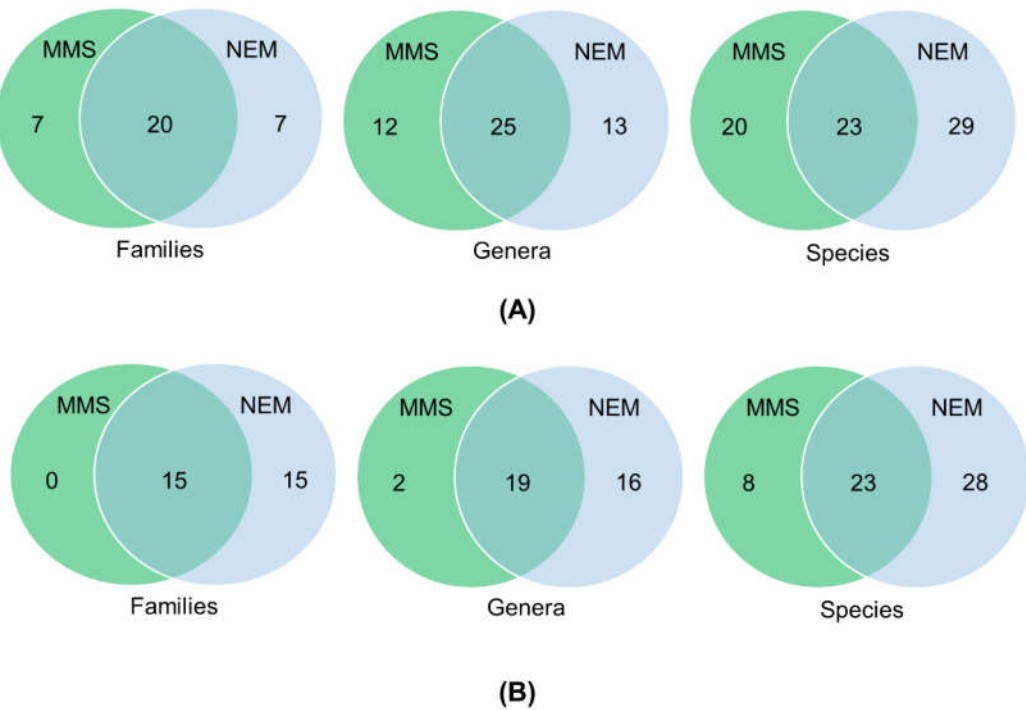

**Figure 2.** Venn diagrams showing the number of nematode taxa detected in soil (**A**) and root samples (**B**) by the MMS (MMSF/MMSR) and NEM (Nemf/18Sr2b) primer sets.

## 4. Discussion

Generally, HTS-based studies of nematode diversity include a nematode extraction step to increase nematode target DNA in the background of DNA from other soil-living organisms due to a lack of nematode-specific primer pairs [21,25–27,48]. Such nematode extraction steps may not be practical in studies of plant–nematode interactions where only small amounts of samples are usually available. We previously developed an amplification strategy for 454 pyrosequencing that selectively amplifies nematode DNA from total soil DNA extractions [35]. In the present study, we modified and adapted this amplification strategy based on the NEM primer set for sequencing on the Illumina MiSeq platform. We compared the NEM primer set with two widely used primer sets and a novel primer set developed in this study for their ability to selectively amplify and detect nematode taxa. In our study, the mock communities were mainly composed of plant-parasitic nematode taxa to test the applicability of the primer sets in detecting and distinguishing between plant-parasitic nematode

taxa. To further explore the suitability and efficiency of the primer sets to study nematode communities including non-parasitic nematode taxa, we tested the primers on mixed communities in a range of soil samples.

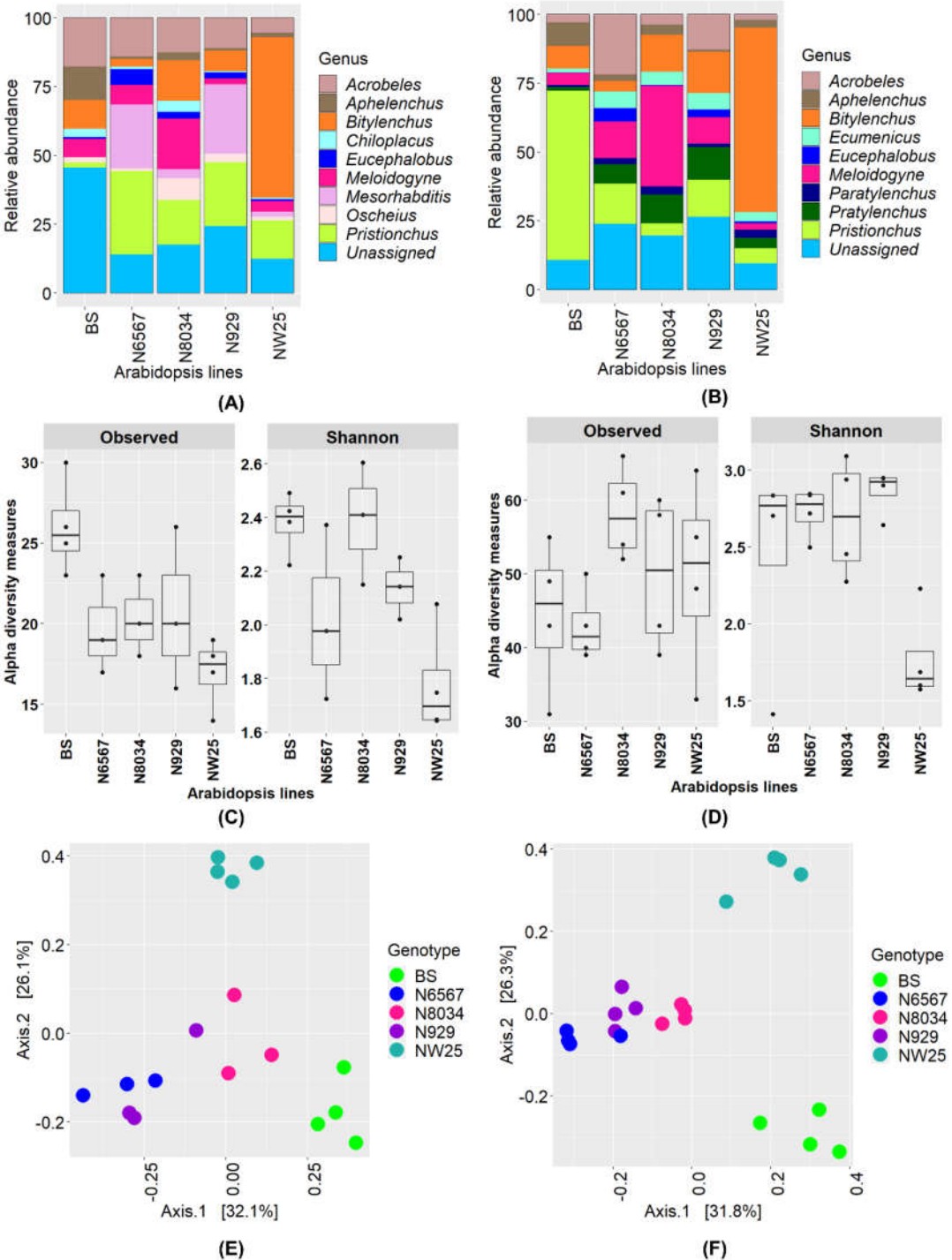

**Figure 3.** Comparison of nematode taxonomic profiles from the roots of different *Arabidopsis* accessions (N6567, N8034, N929, and NW25) and unplanted bulk soil (BS) amplified using MMS: MMSF/MMSR (**A**,**C**, **E**) and NEM: Nemf/18Sr2b (**B**,**D**,**F**) primer sets. (**A**,**B**) Relative abundance of the ten most abundant genera are shown. (**C**,**D**) The alpha diversity was estimated using observed

operational taxonomic unit (out) richness and Shannon diversity indices. (**E,F**) Principal Coordinates Analysis (PCoA) ordination plot using Bray–Curtis distance matrices.

The COI gene-based primer set was initially considered as a suitable marker for barcoding purposes due to its ability to discriminate closely related species in animal phyla [12], as well as its high interspecific and low intraspecific genetic variation in the Nematoda phylum [16]. We found that the COI-targeting JB primer set worked to some extent on extracted nematodes. Hence, JB primers amplified individual nematode species and nematode taxa in mock communities. However, the performance of this primer set using DNA from soil samples was poor; hence, around 95% of the OTUs were not assigned to any taxonomic level. BLAST searches confirmed that these unassigned OTUs did not belong to the Nematoda phylum. Moreover, only four nematode genera and seven species were detected by the JB primer set in soil. As previously suggested by other researchers, the COI gene has high mutation rates and primer sequences are poorly conserved throughout the Nematoda phylum [14,49]. Based on the present study of individual nematode taxa, mock communities, and soil samples without prior nematode extraction, we found that the JB primer set targeting the I3-M11 partition of the COI gene is not suitable for nematode metabarcoding, especially in a soil DNA background.

In a recent study, the small subunit ribosomal RNA gene based SSU primer set (SSU_04F/SSU_22R) outperformed the mitochondrial COI gene based primer set (JB3/JB5GED) for nematode species and genus level detection on extracted nematode samples [21,25]. However, in our study, several agriculturally important nematode taxa were not amplified with the SSU primer set. Furthermore, this primer set resulted in only 1% Nematoda reads from the soil DNA samples. This is supported by earlier studies, where this SSU primer set amplified a large number of non-nematode reads from environmental marine sediment samples [22,50–52]. In the soil samples, the SSU primer set only detected 23 nematode genera compared to the 37 and 38 different genera detected with MMS and NEM primer sets, respectively. Therefore, this primer set is not suitable for targeting nematode diversity without an initial nematode extraction step. The analysis of the individual nematode taxa showed that a better taxonomic resolution was achieved with the MMS primer set that targets the V4-V5 regions of the 18S rRNA gene compared to JB and SSU primer sets. The efficiency of this primer set was further tested using mock communities, in which it was possible to detect all nematode taxa except for *Pratylenchus penetrans*. The MMS primer set detected a high diversity of the nematode communities in the soil samples. However, the primer set detected fewer nematode taxa in the plant root samples compared to the NEM primer set. Additionally, we detected a large number of reads from plants in root samples with the MMS primer set, which further confirmed its poor performance on root samples. This was probably due to competition in primer annealing between nematode and plant DNA templates due to the conserved region in the 18S rRNA gene, even between nematodes and plants.

Most of the nematode taxa in the test of individual nematode species and in the mock communities were amplified and detected using the NEM primer set. This primer set also detected a wide range of nematode taxa in the different soil DNA samples. Furthermore, the NEM primer set amplified 51 nematode species, compared to 31 species by the MMS primer set, in the root samples. Importantly, the NEM primer set efficiently amplified nematode DNA in the presence of plant DNA, thus indicating that this primer set has a higher selectivity in favor of nematode DNA compared to the MMS primer set and is thus promising for root-nematode interaction studies.

The sequence reads from the Rhabditidae family (notably *Mesorhabditis*) were much more relatively abundant in the MMS than in the NEM-generated dataset in the root samples. This discrepancy could have been due to a three-nucleotide mismatch between the 18Sr2b primer of the NEM primer set and the *Mesorhabditis* DNA template. A multiple sequence alignment of 60 different accessions of Rhabditidae (Figure S4) showed that most of the taxa of this family did not show any mismatch except for the three genera (*Mesorhabditis*, *Pelodera*, and *Rhabditis*).

In the present study, different field soils were used to compare the suitability of the metabarcoding primer sets for the detection of nematode diversity in total soil DNA extractions

without prior nematode extraction. For DNA extraction, we only used 0.25 g of soil out of the thoroughly homogenized 100 g subsamples that had been collected under the assumption that the nematode DNA was evenly distributed after the homogenization. If the preservation of diversity should be ensured, we suggest to do several DNA extractions from each soil sample. However, we were still able to detect a large nematode diversity using the NEM primer pair when using this small sample size. The retrieval of a high nematode diversity in only 0.25 g homogenized soil samples shows that the employed metabarcoding strategy is applicable in, e.g., rhizosphere community studies where the total sample weight rarely exceeds a few grams.

## 5. Conclusions

Both the MMS and NEM primer sets outperformed the JB and SSU primer sets based on the nematode detection of individual nematode taxa, mock communities, and soil samples. A comparison of the MMS and NEM primer sets in root samples suggested that the NEM primer set provided a better representation of the nematode community structure in the studied samples. Consequently, we propose the use of the NEM primer set for the detection of nematode taxa on DNA directly extracted from soil, root, or rhizosphere soil samples. An assignment of lower Linnaean taxonomies (genus and species) to sequence reads is a crucial step in the use of DNA markers for biodiversity assessment. Therefore, we conclude that the NEM primer set efficiently detects diverse nematode families and can efficiently detect nematodes at the genus and, in some cases, species ranks in root/rhizosphere soil samples.

**Supplementary Materials:** The following are available online at www.mdpi.com/1424-2818/12/10/388/s1, Figure S1: Location of metabarcoding primers targeting variable regions in mitochondrial (A) and 18S rRNA (B) gene used in the present study. Figure S2. Relative abundance of the nematode genera in soil samples amplified by the JB primer set (A) and SSU primer set (B). Here, unassigned sequence reads were junk DNA, which confirmed by BLAST search. Figure S3. Relative abundance of the nematode genera in soil samples amplified by MMS primer set (A) and NEM primer set (B). Figure S4. Multiple sequence alignment of MMS (A: MMSF) and NEM (B: 18Sr2b) primer sets and representative taxa of Rhabditidae. Table S1: Metabarcoding primer sets used in the present study. Table S2. List of individual nematode species, donating institutes and nematode families of the species used in the study. Table S3. Composition of mock communities used in the study. Table S4. Cropping history and soil properties of twenty different soils used in the study. Table S5. Efficiency of four metabarcoding primers in detection of soil nematodes in twenty different soils at lower taxonomic rank. Table S6. Efficiency of two metabarcoding primers in detection of soil nematodes in roots including background soil. Table S7. Adonis test Permutation analysis of variance (PERMANOVA) for Arabidopsis lines and background soil by MMS primer set. Adonis test was based on Bray–Curtis distance matrices for nematode community dissimilarity assessment using 10,000 permutations. Table S8. Adonis test Permutation analysis of variance (PERMANOVA) for Arabidopsis lines and background soil by NEM primer set. Adonis test was based on Bray–Curtis distance matrices for nematode community dissimilarity assessment using 10,000 permutations.

**Data Availability:** Bioinformatics pipeline, R code, and associated protocols are available from the corresponding author on reasonable request.

**Author Contributions:** Conceptualization, M.M.S., R.S., M.N., M.V., and T.K.; methodology, M.M.S., R.S., and M.N.; software, M.M.S. and R.S.; formal analysis, R.S. and M.M.S.; investigation, M.M.S.; data curation, M.M.S. and R.S.; writing—original draft preparation, M.M.S.; writing—review and editing, M.M.S., M.N., T.K., M.V., and R.S.; visualization, M.M.S.; supervision, M.N., T.K., and M.V.; project administration, M.N., T.K., and M.V.; funding acquisition, M.M.S. and M.N. All authors have read and agreed to the published version of the manuscript.

**Funding:** This research was funded by Aarhus University, Denmark, Project Number: 27747.

**Acknowledgments:** We are grateful to Andrea M. Skantar, United State Department of Agriculture, and Barbara Geric Stare, Sasa Sirca, and Gregor Urek, Department of Plant Protection, Agricultural Institute of Slovenia, for contributing DNA samples of nematode species. We would like to thank Susana Santos for data visualization. We would also like to thank Mathilde Schiøtt Dige and Simone Ena Rasmussen for their excellent laboratory assistance.

**Conflicts of Interest:** The authors declare no conflict of interest.

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
