# Peer review of "Evaluation of Metabarcoding Primers for Analysis of Soil Nematode Communities"

_diversity, doi:10.3390/d12100388_

Round 1
Reviewer 1 Report
The MS is dealing with metabarcoding for analysis of soil nematodes. In general MS is written correctly, however my biggest concern is the usefulness of markers presented in the study on identification of soil nematodes by the use of this method. The whole experiment works great for mock communities but not so great for the samples from the field. The best results was obtained for the primers covering SSU rDNA, which is conservative marker, even if primers covered variable regions. Metabarcoding, by definition should be based on some variable marker (usually COI) and allow to detect even closely related species. Especially for the group of organisms which is rich in species (like nematodes). I am afraid that the use of designed primer set will not have the proper resolution and will not be conclusive in community studies.
Therefore, I am not convinced if the MS should be published. However I do not see the necessity for improvement because the MS is written correctly. I will leave the final decision to the Editor.
Reviewer 2 Report
Sikder et al. evaluated metabarcoding primers for the analysis of soil nematode communities. A number of primers for amplicon sequencing were tested on DNA from nematode taxa, mock communities, soil samples and root samples. Overall they concluded that the Nemf/18Sr2b (a 18S rRNA primer set) had overall the best performance.
After I read the ms carefully I found the information useful for future studies that analyze nematode communities based on amplicon sequencing. The ms is generally well written.
I’ve some questions and remarks regarding the aim to skip the nematode extraction and analyze nematodes directly from the soil:
In line 70-71: The authors suggest that with the method they use, nematode detection is possible without the need for nematode extraction. I agree with that, but I believe the that they should add this is depending on the soil sample (and how many living nematodes are present) and how much soil is used.
In line 221 – 228 (the results) the authors show that with the NEM and MMS primersets a number of nematode taxa were detected. Do I understand it correctly and is this in 0.25 g soil ((see Line 118)? Please add this information.
In line 98 – 101. To evaluate what to expect in the soil samples, please add the details (numbers of living nematodes per 100 g soil) in the text (not only the appendix) about the types of soil samples.
I’m confused about the subsamples of 100 g that were taken, since in L 118 is mentioned that 0.25 g of soil is used for DNA extraction. Could you clarify what is relevant about the 100 g subsample?
In general I believe the authors should be more clear / discuss in more detail / add some additional data about in which situations nematodes in soil (e.g. nematode communities / diversity or abundances or only presence) can be analyzed using DNA extracted from a 0.25 g soil sample.
Reviewer 3 Report
The manuscript is well written.
There are published papers in peer-reviewed international journals, both metabarcoding and molecular taxonomy, that have information on Metabarcoding for analysis of nematode that are not included here.
The abstract is well adjusted and detailed. The introduction is quite poor with respect to the bibliography references. The material and methods section is well described and detailed. In the result section the authors must explain why they only tested the efficiencies of the NEM and MMS primers in roots and a bulk soil samples (page 5, line 230). The discussion section must be improved; their affirmations are poorly argued except in the first and second paragraph of this section.
Round 2
Reviewer 3 Report
Thanks for your corrections!
However, I have some questions:
- D2-D3 region is usefull for identification at species level. Why didn´t you use any primers of the 28S gene? .
- Why did you use only one pair of mitochondrial primer (J3/J5)?. Is this pair of primer better than other mitochondrial primers for amplification the major plant parasitic nematodes?. Explain me, please!. You could suggest future reseach using other DNA regions!
